# Copper-Catalyzed Redox Coupling of Nitroarenes with Sodium Sulfinates

**DOI:** 10.3390/molecules24071407

**Published:** 2019-04-10

**Authors:** Saiwen Liu, Ru Chen, Jin Zhang

**Affiliations:** 1College of Materials and Chemical Engineering, Hunan City University, Yiyang 413000, China; 2Yiyang Agriculture Products Quality Detect Center, Yiyang 413000, China; chenrudaxia@163.com

**Keywords:** copper, redox, additive-free

## Abstract

A simple copper-catalyzed redox coupling of sodium sulfinates and nitroarenes is described. In this process, abundant and stable nitroarenes serve as both the nitrogen sources and oxidants, and sodium sulfinates act as both reactants and reductants. A variety of aromatic sulfonamides were obtained in moderate to good yields with broad substrate scope. No external additive is employed for this kind of transformation.

## 1. Introduction

Nitroarenes are a class of the most readily available starting materials in synthetic organic chemistry. The most important property of nitroarenes is able to undergo various reduction-based reactions, especially for the preparation of primary aromatic amines. The traditional processes mainly rely on using a stoichiometric amount of metal/acid or catalytic hydrogenation [1,2,3,4,5,6]. Direct use of nitroarenes to form new C-N bonds is highly attractive, and great effort has been made on the subject during the past decades [7,8,9]. However, external reducing reagents such as hydrogen and transition-metals are necessary in most cases [10,11,12,13]. Recently, there has been significant interest in the transition-metal-catalyzed C-N bond formation from nitroarenes by use of in situ borrowing-hydrogen strategy [14,15,16,17,18,19]. Among these transformations, two types of reactions can be summarized according to the resultant products: reductive cyclizations and reductive couplings. They provide convenient access to amine, amide and *N*-heterocycles directly from aromatic nitro compounds [20,21,22,23,24,25,26,27,28,29,30,31].

*N*-arylsulfonamides are common building blocks among many pharmaceuticals and biologically active molecules [32,33,34,35]. Various efforts have been made for the synthesis of sulfonamides [36,37,38,39]. The classical methods mainly use arylamines as the nitrogen sources reacting with sulfonyl chlorides or sodium sulfinates [40,41,42,43,44,45]. Alternatively, *N*-arylsulfonamides have been successfully synthesized by the transition-metal-catalyzed cross-coupling of sulfonamides with aryl halides, aryl boronic acids, cyclohexanones, alcohols and hydrocarbons [46,47,48,49,50,51,52,53,54,55,56,57]. Nevertheless, they suffer from some drawbacks of low atom-, step-, and efficiency. In the case of genotoxic and unstable anilines, the undesired impurities may be obtained during the conversion process. Recently, stable and abundant nitroarenes have emerged as a highly attractive nitrogen source for the construction of *N*-arylsulfonamides. A series of Fe-based coupling reaction of sodium arylsulfinates or arylsulfonyl chlorides with nitroarenes to synthesize *N*-arylsulfonamides have developed. This kind of transformation required stoichiometric amounts of NaHSO_3_ or Fe dust as the reductant [58,59,60,61]. Notably, Deng and co-workers developed a palladium-catalyzed method for the synthesis of *N*-arylsulfonamides from arylsulfonyl hydrazides and nitroarenes without external reductants [62,63]. It would be highly desirable to develop a process using inexpensive and non-toxic metal such as copper for the concise and facile preparation of *N*-arylsulfonamides from nitroarenes. We hypothesized sodium sulfinate, which generally served as reducing regent and was oxidized in the sulfonylation reactions, would be participated in the reductive coupling reactions of nitroarenes. Herein, we present a copper-catalyzed reductive coupling of nitroarenes using sodium sulfinate as the coupling partner and reducing regent, which gives rise to an alternative access to pharmacologically significant *N*-aryl sulfonamides (Scheme 1).

## 2. Results

### 2.1. Optimization of Reaction Conditions for Synthesis of 4-Methyl-N-(p-tolyl)benzenesulfonamide ***3a***

To begin our study, the reaction of commercially available *p*-methyl nitrobenzene (**1a**) and sodium 4-methylbenzenesulfinate (**2a**) was chosen as the model under argon at 120 °C to optimize the reaction conditions. Three equivalents of sodium sulfinates were used because the substrate served as a reductant. The product **3a** was obtained in only 10% yield without any metal-catalyst (Table 1, entry 1). The use of FeCl_2_·4H_2_O and FeSO_4_·7H_2_O afforded **3a** in 20% and 11% yields, respectively (entries 2–3). Copper catalyst was found crucial, and various copper salts were investigated for this transformation. Similar results were obtained when employing CuCO_3_·Cu(OH)_2_, Cu(OTf)_2_, CuCl_2_, CuI as the catalyst (entries 4–7). Among the various copper salts examined, CuCl was the most effective, and its use resulted in the formation of **3a** in 92% yield (entry 9). CuBr and Cu powder also promoted the reaction with slightly lower efficiency (entries 8 and 10). The choice of solvents was important for this reaction. The use of DMF and DMSO reduced the reaction yields to 78% and 75%, respectively (entries 11 and 12). Only a trace amount of products were obtained when reactions were carried out in weak polar solvents such as anisole, dioxane, toluene and diglyme (entries 13–16). Unfortunately, a much lower yield was acquired when the reaction was conducted under an atmosphere of air (entry 17).

### 2.2. Substrate Scope for the Nitroarenes

With the optimized conditions in hand, the scope of the reaction with respect to sodium 4-methylbenzenesulfinate (**2a**) and various nitroarenes was investigated (Table 2). The reaction was found to be general, giving the desired *N*-aryl sulfonamides (**3**) in reasonable yields. When nitrobenzene was used, the desired product **3b** was achieved in 70% yield. Common functional substituents such as methoxy and acetyl were compatible with the optimized conditions, and the desired products **3c** and **3d** were obtained in 68% and 70% yields, respectively. Notably, active functional groups such as ester were well tolerated, giving the target products in good yields (**3e**, **3f**). The substituent position on nitroarenes did not show obvious influence on the reaction yields. Moderate yields were obtained when *o*-methylnitrobenzene, *m*-methylnitrobenzene and 2,4-dimethylnitrobenzene were used (**3g**–**3i**). Nitroarenes possessing electron-withdrawing group on the phenyl ring also reacted smoothly with **2a** and afforded the desired product in good yield (**3j**).When naphthalenyl substrate **1k** was used, the desired product **3k** was obtained in 56% yield. To our delight, hetero-nitroarenes such as 6-nitrobenzothiazole and 5-nitroindole also could react with **2a** to give the corresponding sulfonamides in 60% and 42% yields, respectively (**3l**, **3m**).

### 2.3. Substrate Scope for the Sodium Sulfinates

To further examine the scope and the limitations of the reaction, various sodium sulfinates were treated with *p*-methyl nitrobenzene under the standard conditions (Table 3). Sodium benzenesulfinate reacted with **1a** smoothly and gave the desired product **3n** in 81% yield. Other substrates bearing electron-donating group such as methoxy and *tert*-butyl remained effective and gave the corresponding arylsulfonamides in 72% and 53% yields, respectively (**3o**, **3p**). Functional groups halogens were well tolerated and afforded the desired products in moderate yields (entries **3q**–**3s**). The reaction showed poor activity when trifluoromethyl group were presented at the *para* position (**3t**). Gratifyingly, besides aromatic sodium sulfinates, aliphatic sodium methanesulfinate could also react with **1a** to give the target products in 71% yield (**3u**).

### 2.4. Mechanism

To gather more information about the reaction mechanism, a series of control experiments were set up under different reaction conditions (Scheme 2). When the radical scavenger 1,1-diphenylethylene or 2,2,6,6-tetramethyl-piperidine-1-oxyl (TEMPO) was added to the reaction, the desired product **3a** was obtained in 37% and 7% GC yields, respectively (Scheme 2a). This indicates that the present reaction process perhaps wasn’t a free radical mechanism. Treatment of sodium 4-methylbenzenesulfinate **2a** with 4-methyl nitrosobenzene or phenylhydroxylamine under the standard reaction conditions afforded the corresponding product in 32% and 10% yield, respectively (Scheme 2b), which implies both the two compounds were not involved as the key intermediate in this coupling reaction. Notably, no product was detected when *p*-toluidine reacted with sodium sulfinate (Scheme 2c). This means that sodium sulfinate could not directly reduce nitrobenzene into aniline in this catalytic system.

While the exact mechanism of the reductive coupling of nitroarenes with sodium sulfinate is not clear, as the previously reported nitrobenzene-based reduction [61,62,63], these mechanistically experimental results encouraged us to propose an inner pathway of the redox process. As shown in Scheme 3, this reaction starts from the coordination between sodium sulfinate **2** and Cu(I) to form copper(I) sulfinic acid salt **A**. Then, complexation and nucleophilic addition of the lone electron pair of the sulfur moiety to the nitro group of **1** produce a five-membered metallocycle **B**. Next, reduction of **B** by the second molecule of **2** generates complex **C**, followed by the liberation of a sulfonate to afford intermediate **D**. Subsequently, **D** continues to be reduced by the third molecule of **2** to copper *N*-arylsulfonamide salt **E** along with the sulfonate. At last, protonation of **E** affords the desired product and regenerates the Cu catalyst.

## 3. Materials and Methods

### 3.1. General Information

All experiments were carried out under an atmosphere of argon. Flash column chromatography was performed over silica gel 48–75 μm. ^1^H-NMR and ^13^C-NMR spectra were recorded on Bruker-AV (400 and 100 MHz, respectively) instrument (Billerica, MA, USA) internally referenced to SiMe_4_ or chloroform signals. MS analyses were performed on Agilent 5975 GC-MS instrument (EI) (Santa Clara, CA, USA). The structure of known compounds were further corroborated by comparing their ^1^H-NMR, ^13^C-NMR data and MS data with those of literature. All reagents were used as received from commercial sources without further purification. All nitroarenes and sulfinic acid sodium salts **2a**, **2b, 2g 2h**, and **2j** employed were reagent grade materials, and others were prepared according to the literature procedures.

### 3.2. General Procedure for the Preparation of Sodium Sulfinates

4-Methoxybenzenesulfinic acid sodium salt (**2e**) was prepared by heating 2.5 g of sodium sulfite, 2.06 g of 4-methoxybenzenesulphonyl chloride, and 1.68 g of sodium bicarbonate in 9.6 mL of water at 70–80 °C for 4 h. After cooling to room temperature, water was removed under vacuum and the residue was extracted by ethanol, recrystallization as a white solid, the yield was 67% (1.34 g). Similarly, other sodium arenesulfinates were prepared from their corresponding sulfonyl chlorides.

### 3.3. General Procedure for the Synthesis of N-Arylsulfonamides

A pressure tube (10 mL) was charged with CuCl (1.0 mg, 0.01 mmol), *p*-toluenesulfinic acid sodium salt (**2a**, 107.2 mg, 0.6 mmol), *p*-methyl nitrobenzene (**1a**, 27.4 mg, 0.2 mmol) and purged with argon three times. NMP (0.6 mL) and H_2_O (7.2 µL) were added by syringe. The resulting solution was stirred at 120 °C for 40 h. After cooling to room temperature, the crude product mixture was diluted with ethylacetate (15 mL) and washed with a saturated solution of NaCl (3 × 15 mL) and the organic layer was dried over MgSO_4_ and concentrated under reduced pressure. The resulting residue was purified by column chromatography (silica gel, petroleum ether/ethyl acetate = 6:1) to give **3a** as white solid; yield: 40.3 mg (77%) (NMR spectra for all compounds shown in Appendix A).

### 3.4. Product Characterization

*4-Methyl-N-p-tolylbenzenesulfonamide* (**3a**) [58]: White solid, 77% yield (40.3 mg), ^1^H-NMR (400 MHz, CDCl_3_, ppm) δ 7.64 (d, *J* = 8.4 Hz, 2H), 7.24 (d, *J* = 8.0 Hz, 2H), 7.05 (d, *J* = 8.0 Hz, 2H), 6.96 (d, *J* = 8.0 Hz, 2H), 6.51 (s, 1H), 2.40 (s, 3H), 2.29 (s, 3H); ^13^C-NMR (100 MHz, CDCl_3_, ppm) δ 143.7, 136.2, 135.3, 133.9, 129.8, 129.6, 127.3, 122.2, 21.5, 20.8; MS (EI) *m*/*z* (%) 261, 155, 106 (100), 91, 77, 65.

*4-Methyl-N-phenyl-benzenesulfonamide* (**3b**) [58]: White solid, 70% yield (34.6 mg), ^1^H-NMR (400 MHz, CDCl_3_, ppm) δ 7.64 (d, *J* = 7.6 Hz, 2H), 7.26–7.22 (m, 4H), 7.11 (t, *J* = 7.2 Hz, 1H), 7.05 (d, *J* = 7.4 Hz, 2H), 6.55 (s, 1H), 2.37 (s, 3H); ^13^C-NMR (100 MHz, CDCl_3_, ppm) δ 143.9, 136.7, 136.2, 129.7, 129.3, 127.3, 125.2, 121.5, 21.5; MS (EI) *m*/*z* (%) 247, 182, 168, 155, 91 (100), 65.

*4-Methyl-N-(4-methoxyphenyl)-benzenesulfonamide* (**3c**) [58]: White solid, 68% yield (37.7 mg), ^1^H-NMR (400 MHz, CDCl_3_, ppm) δ 7.58 (d, *J* = 7.8 Hz, 2H), 7.24 (d, *J* = 7.5 Hz, 2H), 6.98 (d, *J* = 8.6 Hz, 2H), 6.78 (d, *J* = 8.2 Hz, 2H), 6.20 (s, 1H), 3.78 (s, 3H), 2.41 (s, 3H); ^13^C-NMR (100 MHz, CDCl_3_, ppm) δ 157.9, 143.7, 136.1, 129.6, 129.1, 127.4, 125.3, 114.5, 55.4, 21.5; MS (EI) *m*/*z* (%) 277, 122 (100), 95, 65.

*4-Methyl-N-p-acetylphenyl-benzenesulfonamide* (**3d**) [55]: Off-white solid, 70% yield (44.0 mg), ^1^H-NMR (400 MHz, CDCl_3_, ppm) δ 7.85 (d, *J* = 8.0 Hz, 2H), 7.32 (d, *J* = 7.4 Hz, 2H), 7.26 (s, 2H), 7.14 (d, *J* = 7.7 Hz, 2H), 6.88 (s, 1H), 2.53 (s, 3H), 2.38 (s, 3H); ^13^C-NMR (100 MHz, CDCl_3_, ppm) δ 196.6, 144.4, 141.0, 135.9, 133.4, 129.9, 129.8, 127.2, 119.1, 26.3, 21.5; MS (EI) *m*/*z* (%) 289, 274, 155, 106, 91 (100), 77, 65.

*Methyl-4-(4-methylphenylsulfonamido) benzoate* (**3e**) [58]: Off-white solid, 76% yield (46.4 mg), ^1^H-NMR (400 MHz, CDCl_3_, ppm) δ 7.94 (d, *J* = 7.8 Hz, 2H), 7.72 (d, *J* = 7.6 Hz, 2H), 7.28 (s, 2H), 7.13 (d, *J* = 7.8 Hz, 2H), 6.77 (s, 1H), 3.89 (s, 3H), 2.40 (s, 3H); ^13^C-NMR (100 MHz, CDCl_3_, ppm) δ 166.3, 144.4, 140.9, 132.6, 131.1, 129.8, 127.3, 126.4, 119.2, 52.1, 21.5; MS (EI) *m*/*z* (%) 305, 155, 122, 91 (100), 65.

*Ethyl-4-(4-methylphenylsulfonamido) benzoate* (**3f**) [55]: Off-white solid, 78% yield (49.8 mg), ^1^H-NMR (400 MHz, CDCl_3_, ppm) δ 7.94 (s, 2H), 7.72 (s, 2H), 7.28 (s, 2H), 7.13 (s, 2H), 6.79 (s, 1H), 4.35 (s, 2H), 2.40 (s, 3H), 1.38 (s, 3H); ^13^C-NMR (100 MHz, CDCl_3_, ppm) δ 165.9, 144.4, 140.8, 135.9, 131.1, 129.8, 127.3, 126.8, 119.2, 61.0, 21.5, 14.3; MS (EI) *m*/*z* (%) 319, 274, 155, 119, 108, 91 (100), 65.

*4-Methyl-N-o-tolyl-benzenesulfonamide* (**3g**) [58]: White solid, 55% yield (28.7 mg), ^1^H-NMR (400 MHz, CDCl_3_, ppm) δ 7.60 (d, *J* = 7.7 Hz, 2H), 7.31 (d, *J* = 7.8 Hz, 1H), 7.21 (d, *J* = 7.6 Hz, 2H), 7.14 (s, 1H), 7.08 (s, 2H), 6.23 (s, 1H), 2.39 (s, 3H), 1.99 (s, 3H); ^13^C-NMR (100 MHz, CDCl_3_, ppm) δ 143.8, 136.9, 134.6, 131.5, 130.8, 129.6, 127.6, 126.9, 126.2, 124.4, 21.5, 17.6; MS (EI) *m*/*z* (%) 261, 155, 106 (100), 91, 77, 65.

*4-Methyl-N-m-tolyl-benzenesulfonamide* (**3h**) [58]: White solid, 68% yield (35.5 mg) ^1^H-NMR (400 MHz, CDCl_3_, ppm) δ 7.64 (d, *J* = 7.8 Hz, 2H), 7.22 (d, *J* = 7.6 Hz, 2H), 7.10 (t, *J* = 7.5 Hz, 1H), 6.92 (d, *J* = 7.3 Hz, 1H), 6.88 (s, 1H), 6.83 (d, *J* = 7.5 Hz, 1H), 6.42 (s, 1H), 2.38 (s, 3H), 2.27 (s, 3H); ^13^C-NMR (100 MHz, CDCl_3_, ppm) δ 143.8, 139.3, 136.7, 136.3, 129.6, 129.1, 127.3, 125.9, 122.0, 118.3, 21.5, 21.3; MS (EI) *m*/*z* (%) 261, 196, 182, 155, 106 (100), 91, 77, 65.

*4-Methyl-N-(2,4-dimethylphenyl)-benzenesulfonamide* (**3i**) [64]: White solid, 64% yield (35.2 mg), ^1^H-NMR (400 MHz, CDCl_3_, ppm) δ 7.61 (d, *J* = 7.6 Hz, 2H), 7.24 (d, *J* = 7.5 Hz, 2H), 7.15 (d, *J* = 7.8 Hz, 1H), 6.95 (d, *J* = 7.9 Hz, 1H), 6.92 (s, 1H), 6.18 (s, 1H), 2.41 (s, 3H), 2.28 (s, 3H), 1.96 (s, 3H); ^13^C-NMR (100 MHz, CDCl_3_, ppm) δ 143.6, 137.0, 136.2, 132.3, 131.8, 131.5, 129.6, 127.4, 127.2, 125.3, 21.5, 20.9, 17.6; MS (EI) *m*/*z* (%) 275, 120 (100), 91, 77.

*4-Methyl-N-m-cyanophenyl-benzenesulfonamide* (**3j**) [62]: White solid, 75% yield (40.8 mg), ^1^H-NMR (400 MHz, CDCl_3_, ppm) δ 7.69 (d, *J* = 6.6 Hz, 2H), 7.38–7.35 (m, 3H), 7.29–7.27 (m, 2H), 7.06 (s, 1H), 2.41 (s, 3H); ^13^C-NMR (100 MHz, CDCl_3_, ppm) δ 144.7, 137.8, 135.6, 130.3, 130.0, 128.5, 127.2, 125.1, 123.6, 118.0, 113.4, 21.6; MS (EI) *m*/*z* (%) 272, 155, 91 (100), 65.

*4-Methyl-N-(naphthalene-1-yl)benzenesulfonamide* (**3k**) [62]: White solid, 56% yield (33.3 mg), ^1^H-NMR (400 MHz, CDCl_3_, ppm) δ 7.81 (d, *J* = 5.4 Hz, 2H), 7.72 (s, 1H), 7.62 (d, *J* = 6.4 Hz, 2H), 7.45 (s, 2H), 7.37 (s, 2H), 7.16 (d, *J* = 6.2 Hz, 2H), 6.79 (s, 1H), 2.34 (s, 3H); ^13^C-NMR (100 MHz, CDCl_3_, ppm) δ 143.8, 136.5, 134.3, 131.5, 129.6, 128.9, 128.4, 127.4, 127.2, 126.6, 126.3, 125.4, 122.7, 121.5, 21.5; MS (EI) *m*/*z* (%) 297, 142 (100), 115, 91.

*N-(Benzo[d]thiazol-6-yl)-4-methylbenzenesulfonamide* (**3l**) [65]: Off-white solid, 60% yield (36.5 mg), ^1^H-NMR (400 MHz, CDCl_3_, ppm) δ 8.96 (s, 1H), 7.98 (d, *J* = 8.4 Hz, 1H), 7.84 (s, 1H), 7.67 (d, *J* = 7.3 Hz, 2H), 7.24 (d, *J* = 7.2 Hz, 2H), 7.15 (d, *J* = 8.4 Hz, 1H), 7.03 (s, 1H), 2.39 (s, 3H); ^13^C-NMR (100 MHz, CDCl_3_, ppm) δ 154.0, 150.8, 144.1, 136.0, 134.8, 134.6, 129.8, 127.3, 124.0, 121.0, 114.6, 21.5; MS (EI) *m*/*z* (%) 304, 149 (100), 122, 105, 91, 65.

*N-(1H-Indol-5-yl)-4-methylbenzenesulfonamide* (**3m**): White solid, 42% yield (24.0 mg), ^1^H-NMR (400 MHz, CDCl_3_, ppm) δ 8.18 (s, 1H), 7.59 (d, *J* = 7.3 Hz, 2H), 7.34 (s, 2H), 7.24 (s, 1H), 7.20 (d, *J* = 7.1 Hz, 2H), 6.90 (d, *J* = 8.3 Hz, 1H), 6.49 (s, 1H), 6.31 (s, 1H), 2.38 (s, 3H); ^13^C-NMR (100 MHz, CDCl_3_, ppm) δ 143.4, 136.4, 134.3, 129.4, 128.5, 127.4, 126.3, 125.3, 119.5, 116.7, 111.4, 103.0, 21.5; MS (EI) *m*/*z* (%) 286, 131 (100), 104, 91, 77; HRMS calcd. for C_15_H_15_N_2_O_2_S [M + H]^+^ 287.0784, found 287.0779.

*N-(4-Methylphenyl)-benzenesulfonamide* (**3n**) [58]: White solid, 81% yield (40.0 mg), ^1^H-NMR (400 MHz, CDCl_3_, ppm) δ 7.74 (d, *J* = 7.6 Hz, 2H), 7.54 (t, *J* = 7.2 Hz, 1H), 7.44 (t, *J* = 7.5 Hz, 2H), 7.05 (d, *J* = 7.9 Hz, 2H), 6.94 (d, *J* = 7.9 Hz, 2H), 6.39 (s, 1H), 2.28 (s, 3H); ^13^C-NMR (100 MHz, CDCl_3_, ppm) δ 139.2, 135.5, 133.7, 132.9, 129.9, 129.0, 127.3, 122.5, 20.8; MS (EI) *m*/*z* (%) 247, 106 (100), 77, 51.

*4-Methoxy-N-p-tolylbenzenesulfonamide* (**3o**) [55]: White solid, 72% yield (39.9 mg), ^1^H-NMR (400 MHz, CDCl_3_, ppm) δ 7.69 (d, *J* = 8.5 Hz, 2H), 7.05 (d, *J* = 7.6 Hz, 2H), 6.96 (d, *J* = 7.6 Hz, 2H), 6.90 (d, *J* = 8.4 Hz, 2H), 6.54 (s, 1H), 3.85 (s, 3H), 2.29 (s, 3H); ^13^C-NMR (100 MHz, CDCl_3_, ppm) δ 163.1, 135.3, 133.9, 130.8, 129.8, 129.5, 122.3, 114.2, 55.6, 20.9; MS (EI) *m*/*z* (%) 277, 171, 106 (100), 77.

*4-Tert-butyl-N-p-tolylbenzenesulfonamide* (**3p**) [66]: White solid, yield: 32.1 mg (53%), ^1^H-NMR (400 MHz, CDCl_3_, ppm) δ 7.68 (d, *J* = 7.5 Hz, 2H), 7.45 (d, *J* = 7.4 Hz, 2H), 7.07 (d, *J* = 7.0 Hz, 2H), 6.98 (d, *J* = 6.8 Hz, 2H), 6.39 (s, 1H), 2.30 (s, 3H), 1.32 (s, 9H); ^13^C-NMR (100 MHz, CDCl_3_, ppm) δ 156.7, 136.3, 135.1, 134.0, 129.8, 127.1, 126.0, 122.0, 35.1, 31.1, 20.8; MS (EI) *m*/*z* (%) 303, 182, 133, 106 (100), 77.

*4-Fluoro-N-p-tolylbenzenesulfonamide* (**3q**) [67]: White solid, 50% yield (26.5 mg), ^1^H-NMR (400 MHz, CDCl_3_, ppm) δ 7.75 (t, *J* = 5.9 Hz, 2H), 7.12 (t, *J* = 8.5 Hz, 2H), 7.08 (d, *J* = 7.9 Hz, 2H), 6.95 (d, *J* = 7.8 Hz, 2H), 6.36 (s, 1H), 2.31 (s, 3H); ^13^C-NMR (100 MHz, CDCl_3_, ppm) δ 165.2 (d, *J* = 253 Hz), 135.9, 135.1 (d, *J* = 3.0 Hz), 133.4, 130.1 (d, *J* = 9.4 Hz), 130.0, 122.7, 116.2 (d, *J* = 22.5 Hz), 20.8; MS (EI) *m*/*z* (%) 265, 106 (100), 95, 77.

*4-Chloro-N-p-tolylbenzenesulfonamide* (**3r**) [59]: Off-white solid, 59% yield (33.3 mg), ^1^H-NMR (400 MHz, CDCl_3_, ppm) δ 7.67 (d, *J* = 7.6 Hz, 2H), 7.42 (d, *J* = 7.5 Hz, 2H), 7.08 (d, *J* = 6.7 Hz, 2H), 6.95 (d, *J* = 6.7 Hz, 2H), 6.45 (s, 1H), 2.31 (s, 3H); ^13^C-NMR (100 MHz, CDCl_3_, ppm) δ 139.5, 139.4, 136.2, 133.2, 130.0, 129.3, 128.7, 122.9, 20.9; MS (EI) *m*/*z* (%) 281, 106 (100), 77.

*4-Bromo-N-p-tolylbenzenesulfonamide* (**3s**) [68]: Off-white solid, 63% yield (41.0 mg), ^1^H-NMR (400 MHz, CDCl_3_, ppm) δ 7.59 (s, 4H), 7.08 (d, *J* = 7.3 Hz, 2H), 6.95 (d, *J* = 7.0 Hz, 2H), 6.37 (s, 1H), 2.31 (s, 3H); ^13^C-NMR (100 MHz, CDCl_3_, ppm) δ 138.2, 136.1, 133.2, 132.3, 130.1, 128.8, 128.0, 122.9, 20.9; MS (EI) *m*/*z* (%) 327, 106 (100), 77.

*4-Trifluoromethyl-N-p-tolylbenzenesulfonamide* (**3t**) [55]: White solid, yield: 25.3 mg (40%), ^1^H-NMR (400 MHz, CDCl_3_, ppm) δ 7.87 (d, *J* = 7.1 Hz, 2H), 7.72 (d, *J* = 7.1 Hz, 2H), 7.09 (d, *J* = 6.8 Hz, 2H), 6.97 (d, *J* = 6.7 Hz, 2H), 6.63 (s, 1H), 2.31 (s, 3H); ^13^C-NMR (100 MHz, CDCl_3_, ppm) δ 142.7, 136.4, 134.6 (q, *J* = 33 Hz), 132.9, 130.1, 127.8, 126.1 (q, *J* = 3.7 Hz), 123.2 (q, *J* = 271.1 Hz), 123.0, 20.8; MS (EI) *m*/*z* (%) 315, 145, 106 (100), 77.

*N-p-Tolymethanesulfonamide* (**3u**) [55]: Off-white solid, 71% yield (26.3 mg), ^1^H-NMR (400 MHz, CDCl_3_, ppm) δ 7.16 (d, *J* = 7.2 Hz, 4H), 6.46 (s, 1H), 3.00 (s, 3H), 2.36 (s, 3H); ^13^C-NMR (100 MHz, CDCl_3_, ppm) δ 135.6, 134.1, 130.2, 121.7, 39.0, 20.8; MS (EI) *m*/*z* (%) 185, 106 (100), 79.

## 4. Conclusions

In summary, we have developed a simple synthetic method for the preparation of *N*-arylsulfonamides via copper-catalyzed redox coupling of sodium sulfinates and nitroarenes. Aromatic sulfonamides were formed in moderate to good yields in the absence of external oxidant or reductant. Active functional groups such as ester, cyano, and halogens were well tolerated under the optimized conditions. This method affords an alternative route for the synthesis of *N*-aryl sulfonamides under mild conditions. Moreover, this protocol would inspire other cases of nitroarene reduction-based synthesis of complex compounds. The detailed mechanism and the synthetic applications of this reaction are currently under investigation.

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
