# Peer review of "Copper-Catalyzed Redox Coupling of Nitroarenes with Sodium Sulfinates"

_molecules, 2019, doi:10.3390/molecules24071407_

Round 1
Reviewer 1 Report
Please see the attached file.

Author Response
Response to Reviewer 1 Comments
Point 1: The authors should make clearer why the synthetic method offers a viable alternative to methods reported earlier. The procedure (reported in 2012 by Deng in a patent) is obviously not new. The method uses inexpensive copper, but it requires heating at 120 oC in an argon atmosphere for two days. These conditions do not appear to be attractive for time required or energy expended. Please explain why the method should be further elaborated in the literature.
Response 1: Thank you very much for the constructive comment. This method affords an alternative route for the synthesis of N-aryl sulfonamides under mild conditions. The method uses inexpensive copper. No external additive is employed for this kind of transformation.
Point 2: The method is the same (except the catalyst) as the method of B. Yang et al, (Ref 19, labeled as due to Deng, change to Ding). B. Yang et al use a 4x excess of sulfinate and Pd/C for 12 h. The present paper uses a 3x excess of sulfinate and Cu for 20 h. Both use an Ar atmosphere. Yields are similar. What advantage on a laboratory scale does the method offer over the method of B. Yang?
Response 2: Copper catalyst is cheaper than Pd/C catalyst. Copper is easier to get than palladium.
Point 3: The 2014 patent of X. Yang et al accomplishes the same reaction with the same catalyst as in the present paper at lower temperature in less time, but with a trimethyl borate additive. Why should the method presented in the submitted paper be used instead?
Response 3: No external additive is employed for this kind of transformation.
Point 4: At first glance, the authors’ method of synthesis of sulfonamides appears simple, but the requirement for preparing starting materials (two separate steps) detracts from the efficiency of the method. What advantages are there in having to expend the time and effort of doing these steps?
Response 4: Sodium sulfinates are stable and easy to handle. These compounds have the potential to serve as ideal sulfonyl sources for C-C/C-N bond-forming reactions (Angew. Chem., Int. Ed., 2014, 53, 4205; Org. Lett. 2014, 16, 50).
Point 5: The authors must check to see if compounds reported in the paper have been previously reported. For example, compound 3b has been reported by B. Yang et al (Ref 19). The data in the experimental section do not cite B. Yang et al. All other compounds reported in this paper should be similarly checked for prior reporting.
Response 5: The data in the experimental section have cited related literature. The structure of known compounds were further corroborated by comparing their 1H NMR, 13C NMR and MS data with those of literature. The new compound 3m was characterized by 1H NMR, 13C NMR, MS and HRMS.
Point 6: The introductory paragraphs should be rewritten. For example, “For the diversity synthesis based upon nitroarene reduction, direct use of nitroarenes….” would read better as “Direct use of nitroarenes….” The statement that nitroarenes are used to form new C-N bonds makes no sense because nitroarenes already have C-N bonds. Cu is generally considered to be a transition element.
Response 6: The introductory paragraphs have be rewritten. The words “For the diversity synthesis based upon nitroarene reduction” has been deleted. The new C-N bonds refers to the regenerated C-N bonds, not the C-N bonds of the nitrobenzenes. The word “earth-aboundant” has been replaced with “inexpensive”.
Point 7: Directions should be given for the preparation of commercially unavailable sulfonyl chlorides needed to prepare sulfinates.
Response 7: Sulfonyl chlorides used to prepare sodium sulfinates are commercially available.
Point 8: The experimental procedure specifies “a tube” for carrying out the reaction. The exact nature of the experimental apparatus should be made explicit.
Response 8: The reaction vessel is pressure tube. The words “reaction tube” have been changed into “pressure tube”.
Point 9: Table 1 should show at least one experiment for the use of no Cu catalyst. For example, include an experiment under Ar using NMP or DMSO solvent for two days with 0% yield.
Response 9: The corresponding product 3a was obtained in only 10% GC yield without any metal-catalyst under argon at 120 oC. The use of FeCl2.4H2O and FeSO4.7H2O afforded 3a in 20% and 11% GC yields, respectively. We have added this information to the text (above Table 1).

Reviewer 2 Report
The authors have reported a synthetic method for N-arylsulfonamides via copper-catalyzed redox coupling of sodium sulfinates and nitroarenes. The data is very interesting for N-arylsulfonamides However, the manuscript suffers from several weaknesses:
1) In the optimized reaction condition, the authors used a various copper salts and solvent. How about these reaction without copper catalyst?
2) Any proof to confirm this coupling? Such as products structure information by X-ray diffraction etc.
3) In the experiments about the substrate scope of the nitroarenes and sulfonates, the electronic properties of the groups on the phenyl ring is not enough to be considered.
4) The authors should make the R group clear.
5) The authors mentioned the reaction mechanism of nitroarenes and sodium sulfonates via copper, which is still impossible mechanism and control experiments did not work well for it.
6) The authors used “standard conditions”. “Standard reaction conditions” etc. which is unclear.
Author Response
Point 1: In the optimized reaction condition, the authors used a various copper salts and solvent. How about these reaction without copper catalyst?
Response 1: Thank you very much for the constructive comment. The corresponding product 3a was obtained in only 10% GC yield without any metal-catalyst under argon at 120 oC. This information was added to Table 1.
Point 2: Any proof to confirm this coupling? Such as products structure information by X-ray diffraction etc.
Response 2: The structure of known compounds were further corroborated by comparing their 1H NMR, 13C NMR and MS data with those of literature. The new compound 3m was characterized by 1H NMR, 13C NMR, MS and HRMS. We will upload the supporting information about the 1H and 13C NMR spectra of the products.
Point 3: In the experiments about the substrate scope of the nitroarenes and sulfonates, the electronic properties of the groups on the phenyl ring is not enough to be considered.
Response 3: The electronic properties of the groups on the phenyl ring did not show obvious influence on the reaction yields.
Point 4: The authors should make the R group clear.
Response 4: We have made the corresponding changes. The groups on the phenyl ring of nitroarenes were recorded as R1, and the substituents of sodium sulfinates were recorded as R2.
Point 5: The authors mentioned the reaction mechanism of nitroarenes and sodium sulfonates via copper, which is still impossible mechanism and control experiments did not work well for it.
Response 5: We have done some control experiments under various reaction conditions as refered to some related literatures (for example ref. [17-19]). However, the exact reaction mechanism of nitroarenes and sodium sulfonates via copper is still not clear at this stage. This mechanism can explain that three equivalents of sodium sulfonates were used because the substrate served as a reductant.
Point 6: The authors used “standard conditions”. “Standard reaction conditions” etc. which is unclear.
Response 6: Standard reaction conditions: the mixture of 4-methyl nitrosobenzene or phenylhydroxylamine (0.2 mmol, 1 equiv.), 4-methylbenzenesulfinate sodium (0.6 mmol, 3 equiv.), CuCl (5 mol%), H2O (2 equiv.), NMP (0.6 mL) was stirred under 120 oC for 40 h using an Ar atmosphere.

Reviewer 3 Report
“Copper-Catalyzed Redox Coupling of Nitroarenes with Sodium Sulfinates”
The author described the copper catalyzed coupling of nitroarene with sodium sulfonates. The reaction is very simple and easy to be performed. Although the reaction conditions were similar to the previous Luo’s work (ref 17a), the present reaction would be useful for the synthesis of sulfonamide using nitroarene. Therefore, I recommend that this paper is suitable for the publication in Molecules after major revision.
1. The author should add the data of 1. without the catalyst, 2. with several Fe catalysts that were the optimal catalyst in Luo’s paper and other in ref 7. These data would provide very useful information to the readers.
2. What is the role of 2 equivalent of water? Without water, what happened?
3. The author only used the sodium methanesulfinate as aliphatic substrate. Other aliphatic substrates should be added in substrate scope.
4. In the main text of optimization studies, the author should add a comment that three equivalents of sodium sulfonates were used because the substrate served as a reductant.
5. For the new compounds, IR and HRMS data should be added.
Author Response
Point 1: The author should add the data of 1. without the catalyst, 2. with several Fe catalysts that were the optimal catalyst in Luo’s paper and other in ref 7. These data would provide very useful information to the readers.
Response 1: Thank you very much for the very useful comments. The corresponding product 3a was obtained in only 10% GC yield without any metal-catalyst under argon at 120 oC. The use of FeCl2.4H2O and FeSO4.7H2O afforded 3a in 20% and 11% GC yields, respectively. We have added this information to the text (above Table 1).
Point 2: What is the role of 2 equivalent of water? Without water, what happened?
Response 2: H2O was used as an efficient hydrogen source. The corresponding product 3a was obtained in 75% GC yield when the reaction was carried out in NMP without adding water. This is because the untreated NMP contains a small amount of water. Only a trace amount of product was observed when dried NMP was used without water.
Point 3: The author only used the sodium methanesulfinate as aliphatic substrate. Other aliphatic substrates should be added in substrate scope.
Response 3: We have tried other aliphatic substrates, such as sodium propane-1-sulfinate and sodium cyclopropanesulfinate. The products were obtained in 28% and 19% GC yields, respectively.
Point 4: In the main text of optimization studies, the author should add a comment that three equivalents of sodium sulfonates were used because the substrate served as a reductant.
Response 4: In the main text of optimization studies, We have added a comment that three equivalents of sodium sulfonates were used because the substrate served as a reductant.
Point 5: For the new compounds, IR and HRMS data should be added.
Response 5: There was only one new compound. The new compound 3m was characterized by 1H NMR, 13C NMR, MS and HRMS. We will upload the supporting information about the 1H and 13C NMR spectra of the products.

Round 2
Reviewer 2 Report
The revised manuscript has been improved from its original version, and the authors have addressed all my concerns.